# Autonomic Function Recovery and Physical Activity Levels in Post-COVID-19 Young Adults after Immunization: An Observational Follow-Up Case-Control Study

**DOI:** 10.3390/ijerph20032251

**Published:** 2023-01-27

**Authors:** Ana Paula Coelho Figueira Freire, Shaan Amin, Fabio Santos Lira, Ana Elisa von Ah Morano, Telmo Pereira, Manuel-João Coelho-E-Silva, Armando Caseiro, Diego Giulliano Destro Christofaro, Vanessa Ribeiro Dos Santos, Osmar Marchioto Júnior, Ricardo Aurino Pinho, Bruna Spolador de Alencar Silva

**Affiliations:** 1Department of Health Sciences, Central Washington University, Ellensburg, WA 98926, USA; 2Physiotherapy Department, Universidade do Oeste Paulista (UNOESTE), Presidente Prudente 19050-920, Brazil; 3Exercise and Immunometabolism Research Group, Postgraduate Program in Movement Sciences, Department of Physical Education, Universidade Estadual Paulista (UNESP), Presidente Prudente 19060-900, Brazil; 4Faculty of Sport Science and Physical Education, University of Coimbra, CIDAF, 3000-456 Coimbra, Portugal; 5Polytechnic of Coimbra, ESTESC, 3046-854 Coimbra, Portugal; 6Laboratory for Applied Health Research (LabinSaúde), 3046-854 Coimbra, Portugal; 7Molecular Physical-Chemistry R & D Unit, Faculty of Science and Technology, University of Coimbra, 3004-535 Coimbra, Portugal; 8Postgraduate Program in Movement Sciences, Department of Physical Education, Universidade Estadual Paulista (UNESP), Presidente Prudente 19060-900, Brazil; 9Graduate Program in Health Sciences, School of Medicine, Pontificia Universidade Catolica Do Parana, Curitiba 80215-901, Brazil

**Keywords:** COVID-19, SARS-CoV-2, exercise, autonomic nervous system, sympathetic nervous system, parasympathetic nervous system, COVID-19 vaccination, post-acute sequelae of COVID-19, communicable diseases

## Abstract

Coronavirus disease 2019 (COVID-19) has detrimental multi-system consequences. Symptoms may appear during the acute phase of infection, but the literature on long-term recovery of young adults after mild to moderate infection is lacking. Heart rate variability (HRV) allows for the observation of autonomic nervous system (ANS) modulation post-SARS-CoV-2 infection. Since physical activity (PA) can help improve ANS modulation, investigating factors that can influence HRV outcomes after COVID-19 is essential to advancements in care and intervention strategies. Clinicians may use this research to aid in the development of non-medication interventions. At baseline, 18 control (CT) and 20 post-COVID-19 (PCOV) participants were observed where general anamnesis was performed, followed by HRV and PA assessment. Thus, 10 CT and 7 PCOV subjects returned for follow-up (FU) evaluation 6 weeks after complete immunization (two doses) and assessments were repeated. Over the follow-up period, a decrease in sympathetic (SNS) activity (mean heart rate: *p* = 0.0024, CI = −24.67–−3.26; SNS index: *p* = 0.0068, CI = −2.50–−0.32) and increase in parasympathetic (PNS) activity (mean RR: *p* = 0.0097, CI = 33.72–225.51; PNS index: *p* = 0.0091, CI = −0.20–1.47) were observed. At follow-up, HRV was not different between groups (*p* > 0.05). Additionally, no differences were observed in PA between moments and groups. This study provides evidence of ANS recovery after SARS-CoV-2 insult in young adults over a follow-up period, independent of changes in PA.

## 1. Introduction

The coronavirus disease (COVID-19) is a rapidly spreading condition caused by SARS-CoV-2 virus that infiltrated global populations at alarming rates [1]. Despite severe complications that may arise through contracting COVID-19, a significant number of individuals did not require intensive care (mild to moderate cases) and continued to present with physical [2], neurological [3,4], and other autonomic nervous system (ANS)-related dysfunctions [2,4,5,6,7] four to six months after diagnosis [2,6].

Autonomic modulation was shown to be impaired, even in young adults short after mild and moderate COVID-19 [8,9,10]. The presence of SARS-CoV-2 virus within the carotid body can be a possible mechanism explaining the observed silent hypoxemia and, thus, providing a pathway for nervous system infiltration and subsequent ANS dysfunction [2,8,11,12]. Further, disruptions in autonomic regulation by a viral pathogen are associated with a cytokine storm immune response, which results in oxidative stress leading to cell damage [13].

Cases of COVID-19 among young adults are concerning worldwide since this population can account for 70% of those infected globally [14]. These findings might be attributed to the fact that they are of working age, with high mobility and numerous interpersonal interactions. Although the understanding of ANS impairments after COVID-19 infection in this population is still under investigation, previous research demonstrated early autonomic alterations, including increases in central sympathetic drive and vagal control of the heart [8,9]. ANS changes may be affected by age, body mass, degree of physical activity, and time since diagnosis [15]. Although young adults may present better physiological status and more efficient autonomic regulation when compared to elderly individuals, [16] the effects of COVID-19 on autonomic function over time still need further investigation.

It is well established that ANS plays a major role in modulating homeostasis through influencing systemic bodily functions. Dysfunction of the ANS could result in detrimental effects of downstream physiological processes associated with respiratory, vascular, immune, hematological, and renal processes [17,18]. Increases in SNS activity have also been identified to be an independent predictor of mortality in association with several diseases [19,20]. Considering previous studies that reported persistent symptoms and impairments as a result of post-acute sequelae of COVID-19 (PASC), it is vital to investigate ANS behavior for longer periods of time following infection by SARS-CoV-2 [8,21,22].

Additionally, investigating factors that can influence better ANS outcomes after COVID-19 is essential to advancements in care and intervention strategies. Previous research has found that physical activity (PA) levels can directly influence autonomic modulation [23,24] and can be associated with reduced risk and severity of COVID-19 symptoms [9,21,22]. Concurrently, PA is related to enhancements in immunity and cardiorespiratory fitness and can help to prevent and treat obesity, cardiovascular disease, diabetes, liver disease, cancer, and other chronic diseases [25,26], thus, indirectly reducing the threat of COVID-19 [26]. These findings highlight the magnitude of the role of PA in regulating multiple autonomic system routes to maintain homeostasis and may provide a protective effect from SARS-CoV-2 infection.

An improved understanding of the influences of the long-term effects of COVID-19 on ANS and, consequently, multiple body systems is vital for developing public-health strategies that can reduce the impact of the disease. Clinicians may consider evidence from this literature as a means for guidance to improve clinical assessment and develop non-medication intervention strategies in the prevention and recovery from infection in young adults.

Recently, our group demonstrated the short-term effects of COVID-19 on ANS in young adults [9], but information on long-term effects of COVID-19 still requires research. Young adults are an overlooked population, possibly due to the fact that they are the least at risk for severe negative outcomes. However, the high incidence in young adults [27] emphasizes the need for research into the effects of COVID-19 on this population.

Therefore, the primary aim of this study was to observe the effects of mild to moderate COVID-19 on ANS function over a follow-up period, including before and after six weeks of immunization, in young adults. Secondarily, we aim to identify PA behavior over that period and analyze possible correlations with ANS modulation.

## 2. Materials and Methods

### 2.1. Ethical Approval

The study was conducted based on the ethical standards described in the Declaration of Helsinki. All participants were notified about the study purpose and protocols and provided written informed consent to all protocols. The Ethical Institutional Review Board approved the study (approval number: 38701820.0.0000.5402).

### 2.2. Study Design

This is an observational prospective case-control study, part of a broader project: FIT-COVID Study [28]. The study was previously registered at the Brazilian Clinical Trials Registry (registration number: RBR-5dqvkv3). All reports followed the Strengthening the Reporting of Observational Studies in Epidemiology (STROBE) guidelines [29].

Individuals previously infected with COVID-19 (PCOV) were initially recruited before immunization to participate in the study through local media (television, radio) and social media via electronic access to participants’ database of the Municipal Health Secretariat of Presidente Prudente São Paulo, Brazil (231,953 inhabitants; human development index, 0.806; 23,657 confirmed COVID-19 cases, moving average of 135 cases (May 2021)). An age-matched healthy control group (CT) that was negative for COVID-19 was also recruited. Six weeks after full immunization, individuals in both groups were invited to repeat assessments (Figure 1).

Inclusion criteria were male and female subjects aged 20–40 years with a diagnosis of mild or moderate clinical COVID-19, with a previous positive polymerase chain reaction (PCR) test. Mild cases were considered when patients presented slight clinical symptoms and absence of imaging findings of pneumonia. Moderate cases were considered when patients reported fever or respiratory symptoms [30]. Participants were recruited after a minimum of 15 and a maximum of 120 days of diagnosis by positive PCR test [31]. To screen for confirmed or probable previous SARS-CoV-2 infection for the control group, a lateral flow test for SARS-CoV-2 Immunoglobin G (IgG) and Immunoglobin M (IgM) antibodies was conducted using amplified chemiluminescence and chemiluminescence serological methods, respectively. Subjects in both groups were eligible for a follow-up evaluation if they received any type of SARS-CoV-2 immunization after initial evaluations.

We excluded subjects that presented severe clinical manifestations of COVID-19, including respiratory distress and a respiratory rate > 30 times per minute, fingertip blood oxygen saturation < 93% at rest, and partial arterial oxygen pressure (PaO2)/fraction of inspiration oxygen (FiO2) < 300 mmHg. Subjects that reported critical conditions, including respiratory failure requiring mechanical ventilation, shock, and other organ failure, or requiring intense care treatment, were also excluded [30]. Finally, we also excluded individuals that participated in rehabilitation programs after COVID-19 or that presented any chronic noncommunicable diseases, smokers, history of continous drug use, medications, such as anti-inflammatory drugs, antibiotics, or other drugs known for their impact on the ANS.

### 2.3. Evaluations

Initially, general anamnesis was taken, including sociodemographic characterisitics and self-rated health and medical history (family comorbidities, cardiopulmonary symptoms, and medication use). Participants were then assessed regarding their body mass index (BMI) [32] and PA level [33]. Symptoms that emerged during acute SARS-CoV-2 infection and persistent symptoms were also evaluated [30]. Subjects returned for a follow-up evaluation six weeks after complete SARS-CoV-2 immunization in which assessments were repeated [28]. The number of days between baseline (before immunization) and follow-up (after immunization) assessments was 161.60 ± 45.94 days for CT and 168.42 ± 24.26 days for PCOV (approximately five months).

#### 2.3.1. Body Mass Index

Body mass index (BMI) was calculated according to previous literature [32]. We considered BMI as the ratio of weight (kg) to height (m) squared. Weight was evaluated using an electronic and calibrated scale (Kratos-Cas, São Paulo, Brazil) where subjects wore light clothes and were barefoot. Height was evaluated using a portable anthropometer (Kratos-Cas, São Paulo, Brazil) [32].

#### 2.3.2. Heart Rate Variability

ANS function was measured through heart rate variability (HRV), which is considered a simple, dependable, and noninvasive method [34]. For this evaluation, participants were asked to attend an outpatient clinic and were instructed to arrive in a fasted state, having abstained from exercise, caffeine, chocolate, and alcohol for at least 24 h before evaluation and ≥4 h after a snack or light meal. Evaluations were performed in a silent space, with temperature of ~23 °C. HRV analysis was performed in the morning to avoid extraneous influences of circadian changes [34].

Heart rate was recorded beat to beat to evaluate cardiac autonomic modulation. We used a cardio-frequency meter (Polar RS800CX, Polar Electro, Kempele, Finland) at a 1 kHz sampling rate. Participants were equipped with a chest strap and monitor and remained at rest with spontaneous breathing for 25 min [35]. HRV was performed on 256 consecutive intervals between succesive heartbeats (RR intervals) from the most stable segment on the tachogram. Only series with <5% error were considered suitable for analysis. Kubios HRV^®^ software (Biosignal Analysis and Medical Image Group, Department of Physics, University of Kuopio, Finland) was used to complete the HRV analysis [36,37].

The HRV was assessed in both time and frequency domains using Kubios HRV^®^. For the time domain, the mean RR intervals (reflecting global variability) were used. Additionally, the indexes presented in Table 1 were calculated.

#### 2.3.3. Physical Activity Level

PA level was measured using a triaxial accelerometer (GT3X+; ActiGraph, LLC, Pensacola, FL, USA). Participants were instructed to wear the accelerometer above the waist for seven consecutive days during waking hours. A minimum of four days with at least 10 h per day was considered valid accelerometer data. Participants were instructed not to use the accelerometer while bathing, sleeping, or performing water activities. Every morning, a researcher sent a WhatsApp message reminding the participant to use the accelerometer.

Non-wear periods were defined as time intervals of at least 60 consecutive minutes of zero counts, with an activity interruption allowance of 0–100 counts per minute lasting a maximum of two consecutive minutes [39]. Counts per minute were calculated using the sum of the total activity counts in the vertical axis divided by the valid number of days. Sedentary time was defined as values <100 counts per minute, light PA as values between 100 and 2019 counts per minute, and moderate–vigourous PA (MVPA) as values >2020 counts/minute. Data were processed using ActLife software (version 6.9.2, ActiGraph Pensacola, FL, USA) [33].

#### 2.3.4. Statistical Analysis

Statistical procedures were performed using GraphPad Prism 9 (version 9.3.1; GraphPad Software, Dotmatics, San Diego, CA, USA). Data distribution was analyzed using the Shapiro–Wilk test. For primary analysis of intergroup comparisons between the post-COVID-19 (PCOV) and control groups (CT), the unpaired *t*-test or Mann–Whitney U-test was used according to data distribution. Additionally, the Welch’s correction was used for comparison of variables with different standard deviations. For intragroup paired analysis, paired *t*-test or Wilcoxon test was performed. The Chi-square test was used to assess for differences within the categorical variables of PA. The Fisher exact test was performed to assess for differences between sex distribution.

Effect size (ES) was calculated using the difference between moments to determine the magnitude of differences over time. ES was represented by Cohen’s d values and was classified as d (0.01) = very small, d (0.2) = small, d (0.5) = medium, d (0.8) = large, d (1.2) = very large, and d (2.0) = huge [40].

Secondarily, the data were analyzed using analysis of covariance (ANCOVA) to better understand the influences of PA on the change in HRV indexes [41]. The dependent variable was represented by each HRV index and MVPA was used as the covariate in this multivariable regression. Data were presented with mean and standard deviation or median and respective quartiles (Q1–Q3) according to data distribution. Statistical significance was considered *p* < 0.05.

## 3. Results

In terms of participants, 239 control and 154 post-COVID subjects were assessed for eligibility for this study. Further, 92 control and 33 post-COVID subjects were deemed eligible to participate in this study and 268 individuals did not meet the inclusion criteria. Moreover, 57 of these subjects were evaluated at baseline. After 19 exclusions due to errors in HRV recordings, 38 subjects with complete data were included in the baseline analysis, where 18 subjects were in the CT and 20 in the PCOV group. Subjects then returned six weeks after receiving full COVID-19 immunization. At this stage, 20 participants were excluded due to declining a follow-up assessment and one individual was excluded due to SARS-CoV-2 reinfection. The number of individuals at each stage of the study is presented in Figure 2.

No differences in sex distribution were observed between the groups (*p* = 0.1119). Descriptive characteristics of age, weight, height, and body mass index were uniformly distributed and not different between groups, as indicated in Table 2. Participants in CT returned for follow-up evaluation 161.60 ± 45.94 days after baseline evaluation and PCOV returned after 168.42 ± 24.26 days (approximately five months). In CT, 30% received AstraZeneca immunization, 30% received CoronaVac, and 40% received Pfizer. In PCOV, 57% received CoronaVac immunization and 43% received Pfizer. Additionally, patients in PCOV were evaluated at baseline 50 ± 32.54 days after testing positive for SARS-CoV-2, classified as mild to moderate COVID-19 cases [25].

There were no statistical differences (*p* > 0.05) in PA between CT and PCOV groups (intergroup analysis) and between baseline and follow-up moments (intragroup analysis).

Results for cross-sectional analysis with comparisons between CT and PCOV at baseline are presented in a previous study by the FIT-COVID research group [9].

Figure 3 presents the intragroup analysis. There was no difference between baseline and follow-up moments within CT or PCOV groups (*p* > 0.05) regarding SNS activity (Figure 3). Parasympathetic nervous system (PNS) activity increased between baseline and follow-up moments, reflected by mean RR (*p* = 0.0312) and pNN50 (*p* = 0.0312) for PCOV group (Figure 3). Mean RR increased from 734.50 ± 131.80 ms at the baseline moment to 794.90 ± 105.60 ms at the follow-up moment and pNN50 increased 4.53 ± 4.68% to 11.74 ± 8.81%. No differences between moments were observed in CT (*p* > 0.05). The intragroup analysis between moments also revealed a difference in global variability, reflected by RR triangular index (7.91 ± 2.34 at baseline to 9.60 ± 2.17 at follow-up; *p* = 0.0312; Figure 3).

Table 3 and Appendix A portrays primary analysis of the present study, exhibiting intergroup comparisons between HRV indexes in control (CT) and PCOV groups. (Appendix A). We observed that the PCOV group presents significant reductions in sympathetic activity over time when compared to CT demonstrated by mean HR (*p* = 0.0088) and SNS index (*p* = 0.0068; Table 3; Appendix A). PCOV also presented with a significant increase in PNS activity over time demonstrated by mean RR (−44.54 ± 32.38 vs. 60.36 ± 55.35 ms; *p* = 0.0097) and PNS index (−0.32 ± 0.20 vs. 0.54 ± 0.35; *p* = 0.0091) when compared to CT (Table 3; Appendix A). No intergroup differences were observed in global variability.

Intergroup comparisons between PCOV and CT at the follow-up moment revealed that there were no significant differences between the groups in any HRV index (*p* > 0.05).

ES was determined using Cohen’s d values [40]. Variables that yielded significance in the inter-group analysis had ES reported. Within SNS activity indexes, mean HR (ES = 2.56) and SNS index (ES = 2.56) both had very large ESs. PNS activity indexes of mean RR (ES = 2.31) and PNS index (ES = 3.02) also yielded very large ESs. No global variability indexes yielded significance during the inter-group analysis. Lastly, no PA variables presented significance in either the inter- or intra-group analysis.

The secondary analysis, including MVPA in the multivariable regression analysis model, showed that the significant group differences in mean HR, SNS index, mean RR, and PNS index were maintained (*p* < 0.05) (Table 3).

## 4. Discussion

This study observed the effects of mild to moderate COVID-19 on the ANS in young adults, before and a minimum of six-weeks after complete SARS-CoV-2 immunization. The relationship between PA and the changes in ANS were also analyzed. The primary finding of this study is that autonomic function was improved in young adults who were infected by SARS-CoV-2 after approximately five months of follow-up. This change was characterized by significant decreases in SNS indexes (mean HR and SNS index), while also having significant increases in PNS HRV indexes (mean RR and PNS index). Additionally, when both groups were compared at the follow-up moment, our results revealed that the young adults infected by SARS-CoV-2 presented similar autonomic function when compared to CT. Secondarily, no differences were observed in PA between PCOV and CT groups over the follow-up period. The changes in HRV data were maintained even after statistical adjustments using PA levels.

To our knowledge, this study is the first to observe improvements in autonomic function after mild to moderate post-COVID-19 infection in a young adult population within 164.41 ± 91.47 days. Much of the existing literature focuses on older populations who present with greater risk of negative outcomes [3,4,42] or those infected with severe forms of COVID-19 [8]. Additionally, most of the research focuses on cross-sectional data identifying autonomic dysfunctions associated with post-COVID-19, rather than progression and recovery, making it increasingly difficult to understand ANS modulation over time and the relationship with PA.

Previous work by Freire et al. [9], as part of the cross-sectional data from the Fit-COVID study, revealed the presence of autonomic dysfunction shortly after mild to moderate post-COVID-19 in young adults (represented as baseline comparisons in the present study). The study reported that even in mild and moderate infection, young adults who had COVID-19 had greater sympathetic activity (increased levels of the stress index), decreased parasympathetic activity (lower values of RMSSD and SD1 indices), and global variability (reflected through the SDNN, TINN, and SD2 indexes) when compared to individuals who were not infected. Moreover, in participants who were overweight and obese and/or physically inactive, cardiac autonomic dysregulation was more prominent. [9].

Increases in the SNS are associated with a systemic inflammatory condition characterized by perfusion of inflammatory cytokines in bloodstream and other biomarkers associated with inflammation [3,8,11,13]. Additionally, increases in SNS activity are associated with the secretion of catecholamines, which increases metabolism and cardiac activity, cumulatively increasing cardiac stress [43]. Increases in biomarkers indicating oxidative stress have also been observed in association with COVID-19 and flu-like infections [3,12,44]. The PNS’s influence on anti-inflammatory and restorative processes is also inhibited [45]. Previous studies did not identify the exact mechanism behind inhibition of PNS after COVID-19, but likely a function of the reciprocal nature between the SNS and PNS. Therefore, the increase in oxidative stress and subsequent cytokine storm in tandem with reductions in restorative processes that are associated with SARS-CoV-2 infection could explain the alterations observed at baseline.

It is important to highlight that age plays an important role in ANS regulation. Our study observed young adults without any known chronic diseases. In young adults, protective factors can be related to better ANS regulation and might help explain the responses of ANS after virus infection observed in our study. Healthy young adults can present lower levels of arterial and ventricular stiffening, better myocardial contractility, and preserved organ innervation when compared to elderly individuals [46,47]. Additionally, improved sensitivity of vagal reflexes can lead to greater HRV in this population [16]. Therefore, a combination of these aspects could have influenced the improvements observed in the PCOV group over time. We believe that our results support the use of HRV monitoring as an important, reliable, and accessible method to observe alterations in autonomic function. Deviations in autonomic regulation are related to distancing from homeostasis and are present in multiple diseases, both those that directly afflict the nervous system and those afflicting other organs, where they trigger or enhance pathological symptoms [18,48].

This study did not observe changes in PA over time in either group, nor were PA levels between PCOV and CT different from one another. Although there is a lack of statistical difference to explain observations in this study, which may have been a result of an underpowered sample, PA levels still may have had an influence on the improvements in ANS observed in the PCOV group as median PA levels increased from baseline to follow-up. PA aids in ANS modulation, as the peripheral stress induced by skeletal muscle contraction or mechanical stress on organs sends afferent signals to the central nervous system, thus, inducing catecholamine (epinephrine and norepinephrine) release to alter ANS function to meet metabolic demands of exercise [24]. The release of epinephrine and norepinephrine increases SNS activity, consequently promoting PNS activity during recovery from exercise [49].

The World Health Organization recommends that adults should participate in 150 min of moderate PA per week or 75 min of vigorous PA to observe physiological benefits that can protect against COVID-19 outcomes and reduce the risk of many diseases [50]. At baseline, CT had a median moderate PA of 17.90 min per day (min/day), which equates to 125.30 min per week (min/wk) of moderate PA, while PCOV had a median of 13.36 min/day, equating to 93.52 min/wk of moderate PA. Baseline assessments revealed that CT and PCOV did not meet recommendations for moderate or vigorous PA. These results may be explained by modifications in daily behaviors that were a result of sanitary efforts. Many countries employed lockdowns and restrictions to public areas to avoid the continued spread of COVID-19, which increased barriers to achieve ideal levels of physical activity [51,52].

These barriers were discussed in observations by Huber et. al. [52], in which PA was significantly reduced in young adults during the pandemic. Studies have also found that young adults did not return to pre-pandemic PA levels, which were characterized by a combination of increased sedentary activity and a lower frequency of PA bouts [51]. By the time of the follow-up assessment, PCOV increased moderate PA levels to satisfy PA recommendations from a median of 27.86 min/day (195.02 min/wk) of moderate PA. The increase in PA can likely be explained by the relaxation of social restrictions driven by increasing SARS-CoV-2 immunization rates [53], allowing participants in this study to have easier access to public spaces, such as gyms and community parks.

PA is important in the modulation of immune responses, where increased levels of PA are associated with improved cytokine responses, thus, reducing the severity of COVID-19 disease [54]. PA helps to modulate the release of anti-inflammatory factors and, thus, may aid in immune response to SARS-CoV-2 infection [55]. Specific immune cell activities are regulated by the ANS, where dysfunction in the regulatory system can lead to inadequate immune stimulation [12]. In response, there is an accumulation of reactive nitrogenous species (RNS) and reactive oxygen species (ROS) at the site of cell damage, where persistent inflammation can lead to cellular oxidative damage and damage to the infected host [56]. PA can aid in the immune response to infection through increasing antioxidant capacity and increasing stimulation of the T-helper 2 (TH2) cell pathway [12]. PA increases the level of oxidative stress in skeletal muscle as a product of metabolic processes, where a negative feedback mechanism increases levels of antioxidants as a defense to the presence of ROS and RNS [55,57]. The tight relationship between PA level and immune response illustrates a possible mechanism as to why we observed improved ANS function in this young adult population after SARS-CoV-2 infection [22,54,55,58].

This study presents significant clinical relevance, as it provides further information on the relationship of PA behaviors on recovery after mild to moderate COVID-19 infection in a young adult population. We were able to observe alterations in ANS activity after a five-month follow-up period in a young adult population, highlighted by reductions in SNS activity and increases in PNS activity. Although we hypothesize that this observation could be related to PA levels, our experimental design does not allow us to establish causal effects in this matter. Therefore, we strongly suggest that future studies further investigate this aspect, investigating ANS responses across different ranges of PA, which was not possible in our study due to the small sample size.

This evidence better informs health professionals on non-medicinal approaches to reduce the risk and severity of COVID-19 outcomes by using PA as an alternative medicine. Future research should continue to observe the effects of common population behaviors (which may be unique to age, location of residence, or occupation) on PA levels and the influence of these factors on COVID-19 outcomes. Further research may also look to observe the relationships of frequency, intensity, and duration of PA necessary to improve modulation of cytokine immune responses.

Since we evaluated young adults before and after SARS-CoV-2 immunization, the interpretation of ANS response over time should also consider the potential effects of the vaccine. Although the time point for testing the long-term effects of the vaccine is still relatively short, previous studies comprehensively observed the safety and effectiveness of the COVID-19 vaccine [59]. Reports on the effects of immunization on ANS regulation are still scarce and isolated. Limited case reports previously presented occurrences of post-COVID-19 vaccine-associated postural orthostatic tachycardia syndrome (POTS), a multifactorial and often debilitating type of autonomic dysfunction [60,61,62]. These alterations were observed after immunization (four hours to seven days) and a possible mechanism could be attributed to sinus tachycardia and cerebral hypoperfusion secondary to hypocapnia observed in these cases. This mechanism could result in ischemic hypoxia of the carotid body, leading to chemoreflex activation and sympathetic activation [60]. It is important to highlight that these findings are based on very small samples and the pathophysiological role of the patient’s prior COVID-19 infection still requires further investigation. The reports also state that these observations cannot imply causality of POTS possibly linked to the COVID-19 vaccine [60].

In our study, the aspect of immunization in our study was used as a controlling factor to avoid bias (all individuals were vaccinated upon follow-up). Therefore, the absence of a non-vaccinated group in our design limits our ability to differentiate and account for the specific effects of immunization on ANS. Due to ethical aspects and the high rates of immunization after the availability of immunization in Brazil (over 88% of the population received at least one shot of immunization), it was difficult to incorporate a non-vaccinated group. Future investigations should consider the investigation of ANS function in non-vaccinated individuals to further advance knowledge in this aspect.

Limitations from this study include the loss of follow-up information from baseline to follow-up. CT had 45% and PCOV had 65% loss of participants between observational moments, which may be attributed to the general difficulty in subject compliance for young adults. This high loss of follow-up increases the likelihood of an underpowered sample, which would undermine the statistical findings or explain the lack of statistical differences (50). The effect of the underpowered sample on the statistical findings in this study can only be determined by referencing other studies employing similar methods. Lastly, although we excluded individuals that required intensive care from our sample, subjects may have experienced symptoms, such as chest pain or severe myalgias, indicative of myocarditis or pericarditis, while still not requiring intensive care. This aspect was not controlled in our study and needs to be considered for the interpretation of our results.

Previous research throughout the literature has employed HRV methods that require prolonged monitoring, are subjective in nature, or require expensive equipment that is not accessible to the general population or small-practice clinics. The methods employed in this study are less cumbersome for both healthcare professionals and the patient being monitored, as HRV monitoring can be completed within a single, relatively short visit. Observing changes in ANS activity may be helpful to health professionals to detect signs of infection, track viral progression, and observe dysfunctions caused by the virus, which can aid in the development of individualized intervention strategies that are better informed by HRV data.

Overall, this study observed ANS improvements over a follow-up of an approximately five-month period that may be related to recovery after SARS-CoV-2 infection. To our knowledge, this study is the first of its kind to investigate these outcomes within a four- to six-month period. Lastly, the methods for monitoring ANS function could be used by healthcare professionals to observe the progression of SARS-CoV-2 infection to form individualized non-medication intervention strategies.

## 5. Conclusions

We observed improvements in ANS function after mild to moderate SARS-CoV-2 infection in young adults vaccinated over a five-month period. Additionally, no changes in physical activity levels were observed over the follow-up period.

## Figures and Tables

**Figure 1 ijerph-20-02251-f001:**
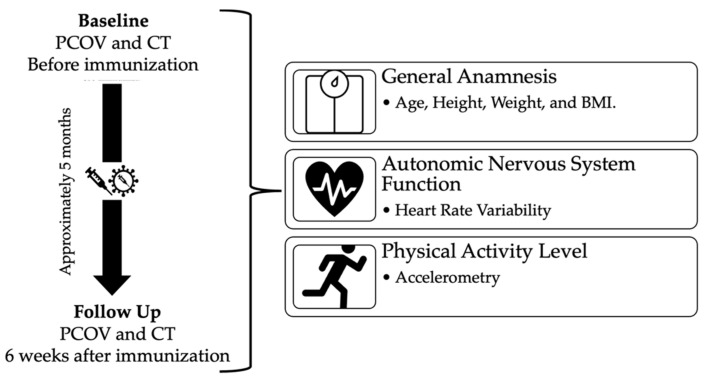
Study design. Evaluations performed on participants at baseline and after a minimum six weeks following complete immunization (follow-up).

**Figure 2 ijerph-20-02251-f002:**
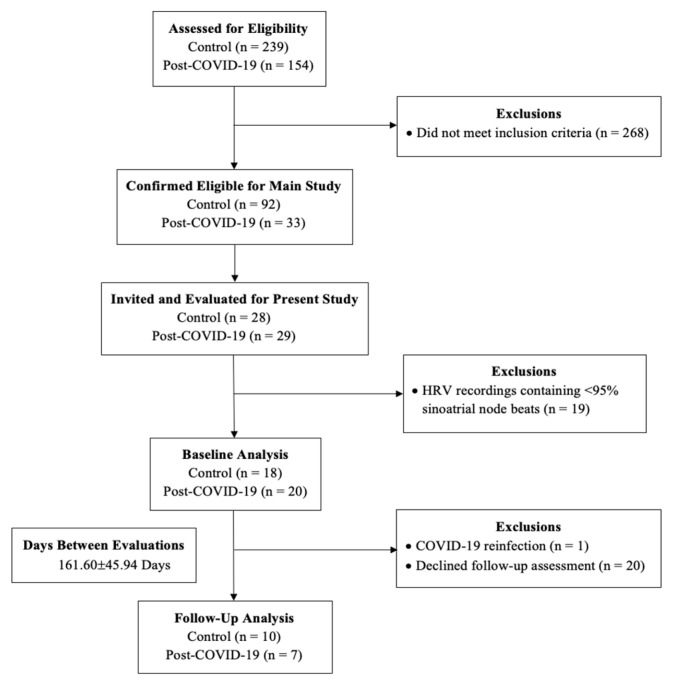
Flow diagram of the study.

**Figure 3 ijerph-20-02251-f003:**
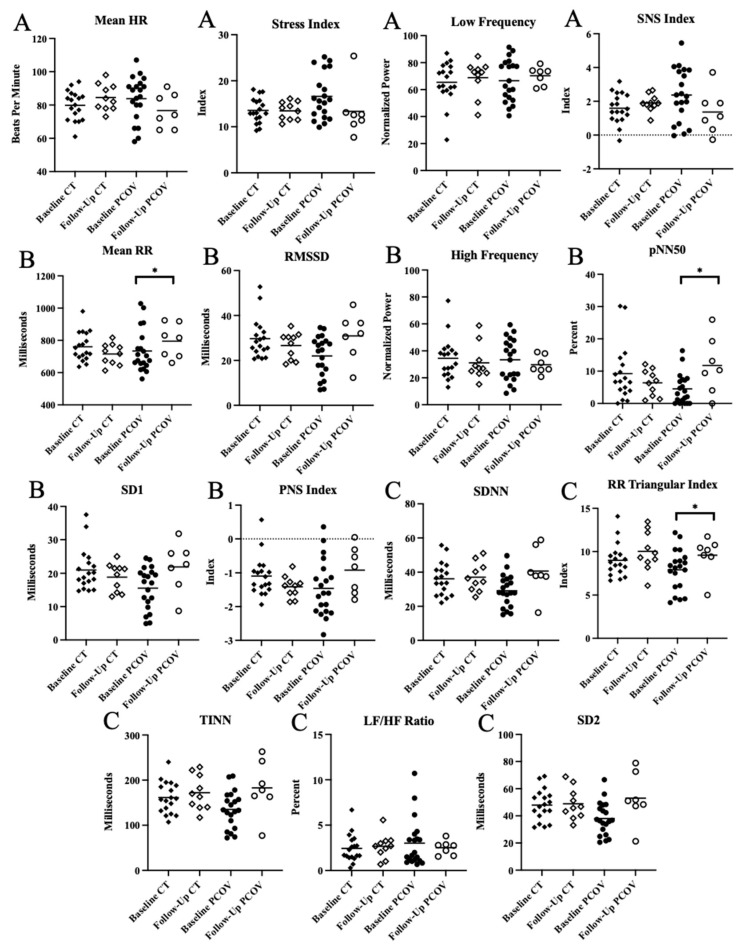
Scatterplot of HRV indexes for paired analysis according to group expressed as the sample mean. (**A**) Sympathetic nervous system activity; (**B**) parasympathetic nervous system activity; (**C**) global variability. * Statistical difference between moments; HR: heart rate; LF: low-frequency component; nu: normalized unit; SNS index: sympathetic nervous system index; RMSSD: root mean square of differences between adjacent normal RR intervals in a time interval; HF: high-frequency component; pNN50: percentage of adjacent RR intervals with a difference in duration >50 ms; SD1: standard deviation of instantaneous beat-to-beat variability; ms: millisecond; PNS index: parasympathetic nervous system index; SDNN: standard deviation of all normal RR intervals recorded in a time interval; RR triangular index: Integral of the density of the RR interval histogram divided by its height; TINN: triangular interpolation of NN interval; LF/HF: ratio of low and high frequency power; SD2: standard deviation of long-term intervals between consecutive heartbeats.

**Table 1 ijerph-20-02251-t001:** Description of heart rate variability (HRV) indexes evaluated. Adapted from Shaffer F et al. [38].

Sympathetic Nervous System Activity	Unit	
Mean HR	bpm	Average heart rate
Stress Index	~	Baevsky’s stress index, a geometric measure of HRV
LF	nu	Relative power between 0.04 Hz to 0.15 Hz
SNS Index	~	Calculated based on the mean HR, Baevsky’s stress index, and SD2 in normalized units
Peripheral Nervous System Activity		
Mean RR	ms	Average time of R-R intervals
RMSSD	ms	Square root of mean squared difference between adjacent RR intervals
HF	nu	Relative power between 0.15 Hz to 0.4 Hz
pNN50	%	Percentage of successive RR intervals that differ by more than 50 ms
SD1	ms	Poincaré plot standard deviation perpendicular the line of identity
PNS index	~	Calculated based on the mean RR, RMSSD, and SD1 in normalized units
Global Variability		
SDNN	ms	Standard deviation of all normal RR intervals
RR Triangular Index	~	Integral of the density of the RR interval histogram divided by its height
TINN	ms	Width of RR interval histogram
LF/HF	%	Ratio of low- and high-frequency power
SD2	ms	Poincaré plot standard deviation along the line of identity

HR: heart rate; LF: low-frequency component; SNS index: sympathetic nervous system index; RMSSD: root mean square of differences between adjacent normal RR intervals in a time interval; HF: high-frequency component; pNN50: percentage of adjacent RR intervals with a difference in duration >50 ms; SD1: standard deviation of instantaneous beat-to-beat variability; SD2: standard deviation of long-term intervals between consecutive heartbeats; PNS index: parasympathetic nervous system index; SDNN: standard deviation of all normal RR intervals recorded in a time interval; TINN: triangular interpolation of NN interval; ms: millisecond; nu: normalized unit; HRV: heart rate variability; Hz: Hertz.

**Table 2 ijerph-20-02251-t002:** Sample characterization and physical activity data.

	Control					Post-COVID						
	Baseline (*n* = 18)		Follow-Up (*n* = 10)		Intragroup Analysis	Baseline (*n* = 20)		Follow-Up (*n* = 7)		Intragroup Analysis	Intergroup Analysis	
	Median	Q1-Q3	Median	Q1–Q3	Difference between Means	Median	Q1-Q3	Median	Q1–Q3	Difference Between Means	95% CI	*p*-Value
Sex (M/F)	13/5	~	8/2	~		9/11	~	4/3	~	~	~	0.1119
Age (years)	26.66	21.11–31.41	~	~	~	28.47	24.73–33.77	~	~	~	~	~
Weight (kg)	71.65	57.85–88.85	62.70	51.20–103.60	−1.67 ± 8.56	77.35	65.75–90.68	67.40	59.70–72.25	-10.31 ± 6.73	-2.20 to 6.25	0.4116
Height (m)	1.76	1.65–1.79	1.69	1.63–1.79	−0.03 ± 0.04	1.71	1.61–1.77	1.70	1.64–1.76	0.004 ± 0.04	−0.02 to 0.01	0.7348
BMI (kg/m^2^)	23.92	21.11–28.20	23.60	20.00–31.98	−0.01 ± 2.29	25.45	23.13–31.34	23.53	19.76–26.41	−3.58 ± 2.47	−0.77 to 2.27	0.2982
Days between Positive Test and Follow-Up	~	~	~	~	~	37.50	24.75–70.50	~	~	~	~	~
Physical Activity												
	*n* = 16		*n* = 8			*n* = 16		*n* = 4				
	Median	IQR	Median	IQR		Median	IQR	Median	IQR			
Sedentary Activity (min/day)	578.43	534.88–676.68	590.52	467.67–664.25	−92.97 ± 98.48	499.59	467.05–593.48	601.15	493.50–646.97	−33.88 ± 146.52	−212.98 to 373.82	0.5550
Light Activity (min/day)	251.50	188.54–291.00	254.14	220.00–407.34	52.77 ± 33.62	285.15	206.75–432.30	258.48	235.24–306.71	−42.21 ± 60.08	−131.75 to 182.60	0.7260
Moderate Activity (min/day)	17.90	5.14–29.87	11.00	6.04–15.57	−10.65 ± 8.58	13.36	9.25–33.72	27.86	23.78–32.25	4.73 ± 11.36	−28.06 to 25.00	0.9004
Vigorous Activity (min/day)	3.65	0.00–15.82	0.00	0.00–2.36	−7.73 ± 4.33	0.00	0.00–0.50	0.13	0.00–4.67	0.92 ± 1.00	−10.63 to 3.26	0.2061
MVPA (min/day)	23.72	12.60–46.00	11.00	6.36–17.13	−19.13 ± 11.86	14.79	9.57–34.04	31.06	24.92–32.72	5.51 ± 11.69	−32.43 to 19.06	0.5756
Steps Count (steps/day)	6570.65	4994.12–7795.24	4822.72	3714.61–8585.47	−1296.60 ± 1706.15	5231.86	3696.86–9601.29	6350.22	5996.22–7197.71	187.06 ± 1585.93	−5935.78 to 4693.48	0.7998
Total Time in BSB 30–60 min Bouts	169.44	97.32–222.25	161.07	63.24–239.47	−18.99 ± 39.95	91.36	73.36–134.18	137.36	98.78–187.97	30.67 ± 35.92	−124.54 to 137.29	0.9157
Total Time in BSB ≥ 60 min Bouts	34.31	19.17–99.61	46.79	4.04–69.32	−17.29 ± 22.46	17.63	8.86–29.32	30.07	2.25–57.36	−4.35 ± 25.22	−66.38 to 77.40	0.8678

Q: quartile range; F: female; M: male; kg: kilogram; m^2^: square meter; BMI: body mass index; MVPA: moderate to vigorous physical activities; BSB: breaks. in sedentary behavior.

**Table 3 ijerph-20-02251-t003:** Comparisons of change in HRV indexes in the post-COVID and control groups and between groups.

	Control (*n* = 11)	Post-COVID (*n* = 7)				
	Intragroup Analysis	Intragroup Analysis	Intergroup Analysis			
	Dif Between Means	Dif Between Means	Unadjusted 95% CI	Unadjusted *p*-Value	Adjusted 95% CI	Adjusted *p*-Value
SNS Activity						
Mean HR	4.67 ± 3.37	−7.13 ± 5.59	−25.83 to −4.33	0.0024 **	−24.67 to −3.26	0.014 **^+^**
Stress Index	−0.10 ± 0.98	−3.26 ± 2.26	−0.60 to 6.51	0.0965	−6.62 to 0.57	0.093
LF (nu)	3.35 ± 5.76	3.66 ± 6.17	−31.16 to 2.17	0.0836	−2.26 to 30.39	0.086
SNS Index	0.31 ± 0.31	−1.01 ± 0.66	−2.48 to −0.29	0.0068 **	−2.50 to −0.32	0.015 **^+^**
PNS Activity						
Mean RR	−44.54 ± 32.38	60.36 ± 55.35 *	33.50 to 222.50	0.0097 **	33.72 to 225.51	0.012 **^+^**
RMSSD	−3.05 ± 3.15	8.95 ± 4.08	−17.06 to 0.80	0.1088	−0.79 to 17.35	0.071
HF (nu)	−3.35 ± 5.76	−3.62 ± 6.17	−2.33 to 31.11	0.0864	−30.33 to 2.40	0.089
pNN50	−2.84 ± 2.93	7.21 ± 2.61 *	−14.62 to 0.19	0.0553	−0.21 to 14.88	0.056
SD1	−2.15 ± 2.24	6.35 ± 2.89	−12.12 to 0.56	0.1038	−0.55 to 12.32	0.07
PNS index	−0.32 ± 0.20	0.54 ± 0.35	0.20 to 1.45	0.0091 **	−0.20 to 1.47	0.013 **^+^**
Global Variability						
SDNN	0.91 ± 3.63	11.54 ± 4.69	−20.92 to 0.96	0.0709	−1.31 to 21.40	0.078
RR Triangular Index	1.02 ± 0.80	1.68 ± 1.01 *	−2.97 to 1.53	0.506	−1.61 to 3.06	0.517
TINN	10.67 ± 14.25	47.86 ± 20.32	−79.60 to 12.72	0.1434	−14.20 to 81.68	0.153
LF/HF	0.23 ± 0.58	−0.50 ± 1.03	−3.00 to 0.75	0.2199	−0.70 to 2.84	0.218
SD2	0.97 ± 4.55	15.03 ± 6.17	−27.16 to 1.65	0.0786	−2.16 to 27.80	0.088

Dif Between Means: difference between means (follow-up—baseline); CI: confidence interval; HR: heart rate; LF: low-frequency component; nu: normalized unit; SNS index: sympathetic nervous system index; RMSSD: root mean square of differences between adjacent normal RR intervals in a time interval; HF: high-frequency component; pNN50: percentage of adjacent RR intervals with a difference in duration > 50 ms; SD1: standard deviation of instantaneous beat-to-beat variability; ms: millisecond; PNS index: parasympathetic nervous system index; SDNN: standard deviation of all normal RR intervals recorded in a time interval; RR triangular index: Integral of the density of the RR interval histogram divided by its height; TINN: triangular interpolation of NN interval; LF/HF: ratio of low and high frequency power; SD2: standard deviation of long-term intervals between consecutive heartbeats. *: Statistical significance (*p* < 0.05) between baseline and follow-up moments. **: Statistical significance (*p* < 0.05) in the difference between means comparing post-COVID-19 and control groups. **^+^**: Statistical significance (*p* < 0.05) of difference between moments between the post-COVID-19 and control groups after ANCOVA according to moderate to vigorous physical activity.

## Data Availability

The data presented in this study are available on request from the corresponding author. The data are not publicly available due to ethical aspects.

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
