# Peer review of "Autonomic Function Recovery and Physical Activity Levels in Post-COVID-19 Young Adults after Immunization: An Observational Follow-Up Case-Control Study"

_ijerph, 2023, doi:10.3390/ijerph20032251_

Round 1

Reviewer 1 Report

This paper from AP Freire et al. investigates the effects of a recent mild to moderate COVID-19 on the ANS function in young adults before and after an average of 5 months from SARS-CoV-2 immunization. Since physical activity can affect ANS balance, the amount of exercise performed by the subjects was considered. Another paper by the Authors (reference number 9, Freire A, et al. Role of Body Mass and Physical Activity in Autonomic Function Modulation on Post-COVID-19 Condition: An Observational Subanalysis of Fit-COVID Study. International journal of environmental research and public health. 2022 ) demonstrated ANS dysfunction after mild to moderate COVID-19 in young adults. The results of that study could be found in the patient baseline characteristics in this study.

The paper found that autonomic function was recovered in young adults who were infected by SARS-CoV-2 after approximately 5 months of follow-up.

The available literature on ANS alterations in subjects who have been affected by COVID-19 (and the role of PA) is now substantial, but many questions have remained unanswered. This paper focuses on an interesting topic and has already been investigated by the Authors. However, some issue needs to be clarified.

Study Design

First of all, no comparison data are provided between those who receive SARS-CoV-2 vaccination and those who don’t in the same period. This is a limitation of the study design. However, the Authors may have some data from the FIT-COVID Study. If not, it would be interesting to show HRV data before immunization, but some months after COVID-19 recovery. Alternatively, the Author should discuss the literature on this topic, and provide a better explanation for this study design.

If the Authors will be able to provide this information and discuss it, there are some issues to be changed or clarified that will be discussed below.

Throughout the paper, the control group is abbreviated CT, or CG; or the abbreviation is twice (or more) declared (see Methods). Please be consistent and revise all the text.

The abstract summarizes well the aim and the results of the paper.

-       I suggest changing the sentence in lines 28 and 29: “Additionally, physical activity (PA) helps improve ANS modulation, where investigation of PA influence on ANS recovery is vital to reduce risk and severity of symptoms”. It is ambiguous and might lead to the conclusion that physical activity could have a fundamental effect on the severity of COVID-19

-       Line 31: correct the word “amnamnesis

The introduction explains the magnitude of ANS dysfunction in COVID-19 and other several diseases. However, only a few sentences are dedicated to this issue in young adults and the health concerns for this population.

1.     The long-term health effects of SNA dysfunction in young adults after COVID-19 are unknown. However, it is unlikely that they could be compared with SNA dysfunction in chronic disease. Please go into detail about the possible burden of SNA dysfunction in young adults.

2.     From line 67 to line 73 the role of physical activity in SNA modulation and in regulating the severity of COVID-19 is described. Apart from its role in sympathetic modulation, physical activity can improve the presentation of COVID-19 because it acts on known risk factors (such as hypertension and systemic inflammation). Please explain better this difference.

3.     Another issue that should be introduced is the possible role of SARS-CoV-2 immunization and ANS balance. This is one of the primary aims of the paper

4.     Furthermore, below I point out a series of mistakes in the text that should be reviewed.

-       Line 59: change “could results” to “could result”

-       Line 61: change “also been indentified to” to “also been identified to

-       Line 62: change “as a results” to “as results”

-       Line 72: add a subject after “by which”, or rewrite the sentence

-       Line 74: change “long term” to “long-term

-       Line 80: change “short term” to“short-term

-       Line 81: change “still needs” to “still need”, or rewrite the sentence with the proper subject

Methods (up to PA levels, included)

The study design and methods are well described. In line  167 I suggest the Authors change “Adapted from (33)”, to “Adapted from Shaffer F et al. (33)”. In addition:

-       Line 106: change “inhabitiants”, with “inhabitants

-       Line 137: correct the decimal place of the numbers

Methods (Statistical Analysis), Table 2 and Results

In lines 203 and 204, the values of Cohen’s d declared do not correspond to the values from Cohen and Sawilowsky (Cohen, Jacob Statistical Power Analysis for the Behavioral Sciences (1988); Sawilowsky, Shlomo S. (2009) "New Effect Size Rules of Thumb," Journal of Modern Applied Statistical Method). Furthermore, also the reference indicated does not report the values declared. Please, correct and report the right values.

Table 2 should be completely rewritten. There are some important mistakes, such as cell exchanges (for example weight with height); the M/F proportions do not match the whole number of the group at the different phases. Some variables (e.g. Moderate Activity) have different decimal digits among columns. For continuous variables, the Authors reported the median and 25th and 75th percentiles. These two quartiles (Q1 & Q3) have been incorrectly called IRQ (which is the difference between Q3 and Q1). I suggest showing parametric data with mean standard deviation and using the median with 25th – 75th percentiles (or IRQ) only for non-parametric data. I would like to point out to the Authors that the median is given in Table 2 and the mean is shown in the text. If the Authors want to use the median, please be consistent in the text and explain why in the Methods section. Furthermore, in the bottom caption of Table 2, IQR and MVPA are repeated (line 238). Please rewrite Table 2 and its caption. I suggest marking the intra-group and inter-group significant differences to make the results more understandable.

In the caption of Table 3 (lines 281-284) “between” and “difference” are repeated twice. Please correct.

Discussion

Please briefly report data of the Authors from Freire et al. (9) in the Discussion Section (especially between lines 317 and 326) to make the results of this paper clearer.

Line 369-370: “PA helps to modulate the release of anti-inflammatory factors, and thus may aid in immune response to SARS-CoV-2 infection”. Line 379-381: “The tight relationship between PA level and immune response illustrates a possible mechanism as to why we observed improved ANS function in this young adult population after SARS-CoV-2 infection.” The interaction between PA and COVID-19 is explained between lines 367 and 381 refer to paper 47, 12, 48, 49. Among these 4 paper, only one (47, Mohamed AA et al) explore the role of PA and COVID-19 and it has a sample size of 30 patients (Intervention: 15, Controls: 15): the other 3 papers does not refer to SARS-CoV-2 infection. Please add specific references (PA and COVID-19) to empower the statements in lines 369/370 and 379/381, or change the discussion from line 367 to line 381.

Line 386-388: “This observation is likely due to the PA level of this population as CT and PCOV met CDC recommendations for physical activity by the time of the follow-up assessment”. Based on the paper data the Authors could not state this. To demonstrate this causality a proper experiment needs to be run, comparing time without MVPA and time plus MVPA. Please change this sentence, or prove it with consistent literature.

Line 394-397: “Additionally, future studies may also aim to observe how behavioral changes resulting from the COVID-19 395 pandemic have impacted PA behavior and further explore how this may have impact ANS function.” This sentence should be rewritten better.

Line 406-408: “ANS dysregulation indicates deviation from homeostasis and can effectively predict cardiometabolic illness which is directly correlated with mortality in several diseases, regardless of age (15)”. This sentence is too vague and the reference discusses mainly cardiac disease. Please be more specific in discussing ANS dysfunction and focus on young adults.

Line 424-426: “The costs associated with equipment and the time for the HRV monitoring in this study are more accessible than other methods, which may work to improve health outcomes after COVID-19” Add references to sustain this statement.

As stated above, Discussion does not explain the implications of SARS-CoV-2 immunization for ANS balance

In addition, I report below some grammatical errors in the Discussion and Conclusion sections that should be reviewed.

-       Line 298: change “6-weeks” in “6 weeks”

-       Lines 301,304, 430: change “follow up” to “follow-up”

-       Line 317: change “cross sectional” to “cross-sectional”

-       Line 343: change “increase” to “increases”

-       Line 405: “as evidence as to the use”. Change prepositions to make clearer the whole sentence

-       Line 412: change “cumbersome on both” to “cumbersome for both”

-       Line 422: change “of monitoring of ANS” to “of monitoring ANS”

-       Line 429: change “5 month” to 5-month”

Author Response

REVIEWER 1

Comments and Suggestions for Authors

This paper from AP Freire et al. investigates the effects of a recent mild to moderate COVID-19 on the ANS function in young adults before and after an average of 5 months from SARS-CoV-2 immunization. Since physical activity can affect ANS balance, the amount of exercise performed by the subjects was considered. Another paper by the Authors (reference number 9, Freire A, et al. Role of Body Mass and Physical Activity in Autonomic Function Modulation on Post-COVID-19 Condition: An Observational Subanalysis of Fit-COVID Study. International journal of environmental research and public health. 2022 ) demonstrated ANS dysfunction after mild to moderate COVID-19 in young adults. The results of that study could be found in the patient baseline characteristics in this study.

The paper found that autonomic function was recovered in young adults who were infected by SARS-CoV-2 after approximately 5 months of follow-up.

The available literature on ANS alterations in subjects who have been affected by COVID-19 (and the role of PA) is now substantial, but many questions have remained unanswered. This paper focuses on an interesting topic and has already been investigated by the Authors. However, some issue needs to be clarified.

Study Design

  1. First of all, no comparison data are provided between those who receive SARS-CoV-2 vaccination and those who don’t in the same period. This is a limitation of the study design. However, the Authors may have some data from the FIT-COVID Study. If not, it would be interesting to show HRV data before immunization, but some months after COVID-19 recovery. Alternatively, the Author should discuss the literature on this topic, and provide a better explanation for this study design.

Answer:  Thank you for your comment. We agree with the comment. However, the aspect of immunization in our study was used as a controlling factor to avoid bias. To better clarify our study design, our initial evaluations started right before SARS-CoV-2 immunization was available in Brazil. Therefore, baseline data for all individuals is before immunization (please, see Figure 1) and we restricted the follow-up analysis to individuals that were vaccinated, attempting to minimize possible cofounders (mix of vaccinated and non-vaccinated on follow-up analysis).

Due to ethical aspects and the high rates of immunization after vaccines availability (over 88% of the population received at least one shot of immunization), we believe it would be difficult to incorporate a non-vaccinated group.

Nevertheless, our main aim was to investigate ANS behavior after COVID-19 and not necessarily the effects of immunization itself on this aspect (which we agree would require a non-vaccinated group). We also standardized the timeframe for evaluation after immunization (at least 6 weeks) for both groups. Therefore, our design allows us to identify ANS responses over that period, but we are limited to differentiating any causal effects of SARS-CoV-2 immunization in this process due to the absence of a non-vaccinated group. We agree that this is important information to be included in the manuscript. Please, see highlighted modifications on the manuscript.

Discussion, Paragraph 12

“Since we evaluated young adults before and after SARS-CoV-2 immunization, the interpretation of ANS response overtime should also consider the potential effects of the vaccine. Although the time point for testing long-term effect of the vaccine is still relatively short, previous studies comprehensively observed the safety and effectiveness of the COVID‐19 vaccine. (59) Reports on the effects of immunization on ANS regulation are still scarce and isolated. Limited case reports previously presented occurrences of post-COVID-19-vaccine-associated postural orthostatic tachycardia syndrome (POTS), a multifactorial and often debilitating type of autonomic dysfunction. (60-62) These alterations were observed short-term after immunization (4 hours to 7 days) and a possible mechanism could be attributed to sinus tachycardia and cerebral hypoperfusion secondary to hypocapnia observed in these cases. This mechanism could result in ischemic hypoxia of the carotid body, leading to chemoreflex activation and sympathetic activation. (60) It is important to highlight that these findings are based on very small samples and the pathophysiological role of the patient’s prior COVID-19 infection still requires further investigation. The reports also state that these observations cannot imply causality of POTS possibly linked to the COVID-19 vaccine. (60)

In our study, the aspect of immunization in our study was used as a controlling factor to avoid bias (all individuals were vaccinated on follow-up). Therefore, the absence of a non-vaccinated group in our design limits our ability to differentiate and account for the specific effects of immunization on ANS. Due to ethical aspects and the high rates of immunization after the availability of immunization in Brazil (over 88% of the population received at least one shot of immunization), it was difficult to incorporate a non-vaccinated group. Future investigations should consider the investigation of ANS function in non-vaccinated individuals to further advance knowledge in this aspect.”

  1. If the Authors will be able to provide this information and discuss it, there are some issues to be changed or clarified that will be discussed below.

Answer:  Thank you for your comment. Please see the previous comment that addresses this point.

  1. Throughout the paper, the control group is abbreviated CT, or CG; or the abbreviation is twice (or more) declared (see Methods). Please be consistent and revise all the text.

Answer:  Thank you for your comment. We reviewed this aspect and included CT as an abbreviation across the manuscript. Please, see highlighted modifications on the manuscript.

  1. The abstractsummarizes well the aim and the results of the paper. I suggest changing the sentence in lines 28 and 29: “Additionally, physical activity (PA) helps improve ANS modulation, where investigation of PA influence on ANS recovery is vital to reduce risk and severity of symptoms”. It is ambiguous and might lead to the conclusion that physical activity could have a fundamental effect on the severity of COVID-19.

Answer:  Thank you for your comment. We reviewed this aspect and changed this sentence in the abstract. Please, see highlighted modifications on the manuscript.

Abstract

“Since physical activity (PA) can help improve ANS modulation, investigating factors that can influence HRV outcomes after COVID-19 is essential to advancements in care and intervention strategies.”

  1. Line 31: correct the word “amnamnesis

Answer:  Thank you for your comment. We reviewed this aspect and changed this word on the abstract. Please, see highlighted modifications on the manuscript.

  1. The introductionexplains the magnitude of ANS dysfunction in COVID-19 and other several diseases. However, only a few sentences are dedicated to this issue in young adults and the health concerns for this population. The long-term health effects of SNA dysfunction in young adults after COVID-19 are unknown. However, it is unlikely that they could be compared with SNA dysfunction in chronic disease. Please go into detail about the possible burden of SNA dysfunction in young adults.

Answer:  Thank you for your comment. A paragraph addressing alternations of ANS in young adults was added in the introduction. Please, see highlighted modifications on the manuscript.

Introduction, Paragraph 4

“Cases of COVID-19 among young adults are concerning worldwide since this population can account for 70% of those infected globally. (20) These findings might be attributed to the fact that they are of working age with high mobility and numerous interpersonal interactions. Although the understanding of impairments of ANS in this population is still under investigation, previous research demonstrated early autonomic alterations, including increases in central sympathetic drive and vagal control of the heart. (8, 9). ANS changes may be affected by age, body mass, degree of physical activity, and time since diagnosis. (21) Although, young adults may present better physiological status and more efficient autonomic regulation,(22) the effects of COVID-19 on autonomic function over time still need further investigation.”

  1. From line 67 to line 73 the role of physical activity in SNA modulation and in regulating the severity of COVID-19 is described. Apart from its role in sympathetic modulation, physical activity can improve the presentation of COVID-19 because it acts on known risk factors (such as hypertension and systemic inflammation). Please explain better this difference.

Answer:  Thank you for your comment. We agree with the comment. Please, see highlighted modifications on the manuscript.

Introduction, Paragraph 5

“Concurrently, PA is related to enhances on immunity, and cardiorespiratory fitness and helps prevent and treat obesity, cardiovascular disease, diabetes, liver disease, cancer, and other chronic diseases(24, 25), thus, indirectly reducing the threat of COVID-19. (25) These findings highlight the magnitude of PA role in regulating multiple autonomic systems routes to maintain homeostasis and may provide a protective effect from symptoms experienced from SARS-CoV-2 infection..”

  1. Another issue that should be introduced is the possible role of SARS-CoV-2 immunization and ANS balance. This is one of the primary aims of the paper.

Answer:  Thank you for your comment. As previously discussed in comment 1, the aspect of immunization in our study was used as a controlling factor to avoid bias. The absence of a non-vaccinated group limits our ability to make conclusions regarding the effects of immunization. We hope that the additional information included in the methods and discussion sections would be sufficient to clarify this aspect.

  1. Furthermore, below I point out a series of mistakes in the text that should be reviewed.

-       Line 59: change “could results” to “could result”

-       Line 61: change “also been indentified to” to “also been identified to

-       Line 62: change “as a results” to “as results”

-       Line 72: add a subject after “by which”, or rewrite the sentence

-       Line 74: change “long term” to “long-term

-       Line 80: change “short term” to“short-term

-       Line 81: change “still needs” to “still need”, or rewrite the sentence with the proper subject

Answer:  Thank you for your comment. We agree with the comment. Please, see highlighted modifications on the manuscript.

  1. Methods (up to PA levels, included)

The study design and methods are well described. In line  167 I suggest the Authors change “Adapted from (33)”, to “Adapted from Shaffer F et al. (33)”. In addition:

-       Line 106: change “inhabitiants”, with “inhabitants

-       Line 137: correct the decimal place of the numbers

Answer:  Thank you for your comment. We agree with the comment. Please, see highlighted modifications on the manuscript.

  1. Methods (Statistical Analysis), Table 2 and Results

In lines 203 and 204, the values of Cohen’s d declared do not correspond to the values from Cohen and Sawilowsky (Cohen, Jacob Statistical Power Analysis for the Behavioral Sciences (1988); Sawilowsky, Shlomo S. (2009) "New Effect Size Rules of Thumb," Journal of Modern Applied Statistical Method). Furthermore, also the reference indicated does not report the values declared. Please, correct and report the right values.

Answer:  Thank you for your comment. We replaced the reference and updated the effect sizes classification. Please, see highlighted modifications on the manuscript.

Statistical analysis, Paragraph 2

“Effect size (ES) was calculated using the difference between moments to determine the magnitude of differences over time. ES was represented by Cohen’s d values and was classified as d (.01) = very small, d (.2) = small, d (.5) = medium, d (.8) = large, d (1.2) = very large, and d (2.0) = huge.(39)”

  1. Table 2 should be completely rewritten. There are some important mistakes, such as cell exchanges (for example weight with height); the M/F proportions do not match the whole number of the group at the different phases.

Answer:  Thank you for your comment. We corrected these mistakes. Please, see highlighted modifications on the manuscript.

  1. Some variables (e.g. Moderate Activity) have different decimal digits among columns.

Answer: Thank you for your comment. We corrected these mistakes. Please, see highlighted modifications on the manuscript.

  1. For continuous variables, the Authors reported the median and 25th and 75th percentiles. These two quartiles (Q1 & Q3) have been incorrectly called IRQ (which is the difference between Q3 and Q1). I suggest showing parametric data with mean standard deviation and using the median with 25th – 75th percentiles (or IRQ) only for non-parametric data.

Answer: Thank you for your comment. We corrected these mistakes. Since the majority of data in this table followed nonparametric distribution we presented these results with median and their respective Q1 and Q3.  Please, see highlighted modifications on the manuscript.

  1. I would like to point out to the Authors that the median is given in Table 2 and the mean is shown in the text. If the Authors want to use the median, please be consistent in the text and explain why in the Methods section.

Answer: Thank you for your comment. The data presented in means in the results are related to HRV data that majorly presented normal distribution and was presented in mean and standard deviation (text and table). Further explanations of data presentation were provided in the statistical analysis section. Please, see highlighted modifications on the manuscript.

  1. Furthermore, in the bottom caption of Table 2, IQR and MVPA are repeated (line 238). Please rewrite Table 2 and its caption. I suggest marking the intra-group and inter-group significant differences to make the results more understandable.

Answer: Thank you for your comment. Since no statistical differences were observed in Table 2 (intra or inter-groups), no marks were necessary.

17.In the caption of Table 3 (lines 281-284) “between” and “difference” are repeated twice. Please correct.

Answer:  Thank you for your comment. We corrected these mistakes. Please, see highlighted modifications on the manuscript.

 Discussion

  1. Please briefly report data of the Authors from Freire et al. (9) in the Discussion Section (especially between lines 317 and 326) to make the results of this paper clearer.

Answer:  Thank you for your comment. Please, see highlighted modifications on the manuscript.

Discussion, Paragraph 3

“Previous work by Freire et al. (9) as part of the cross-sectional data from the Fit-COVID study revealed the presence of autonomic dysfunction short after mild to moderate post-COVID-19  in young adults (represented as baseline comparisons in the present study). The study reported that even in mild and moderate infection, young adults who had COVID-19 had greater sympathetic activity (increased levels of the stress index), decreased parasympathetic activity (lower values of RMSSD and SD1 indices), and global variability (reflected through the SDNN, TINN, and SD2 indexes) when compared to uninfected individuals. Moreover, in participants who were overweight and obese and/or physically inactive, cardiac autonomic dysregulation was more prominent. (9).”

  1. Line 369-370: “PA helps to modulate the release of anti-inflammatory factors, and thus may aid in immune response to SARS-CoV-2 infection”. Line 379-381: “The tight relationship between PA level and immune response illustrates a possible mechanism as to why we observed improved ANS function in this young adult population after SARS-CoV-2 infection.” The interaction between PA and COVID-19 is explained between lines 367 and 381 refer to paper 47, 12, 48, 49. Among these 4 paper, only one (47, Mohamed AA et al) explore the role of PA and COVID-19 and it has a sample size of 30 patients (Intervention: 15, Controls: 15): the other 3 papers does not refer to SARS-CoV-2 infection. Please add specific references (PA and COVID-19) to empower the statements in lines 369/370 and 379/381, or change the discussion from line 367 to line 381.

Answer:  Thank you for your comment. The following references were included to support the information in this paragraph.

  1. Després JP. Severe COVID-19 outcomes - the role of physical activity. Nat Rev Endocrinol. 2021;17(8):451-2.
  2. Mohamed AA, Alawna M. The effect of aerobic exercise on immune biomarkers and symptoms severity and progression in patients with COVID-19: A randomized control trial. J Bodyw Mov Ther. 2021;28:425-32.
  3. Gualano B, Lemes IR, Silva RP, Pinto AJ, Mazzolani BC, Smaira FI, et al. Association between physical activity and immunogenicity of an inactivated virus vaccine against SARS-CoV-2 in patients with autoimmune rheumatic diseases. Brain Behav Immun. 2022;101:49-56.
  4. Yang J, Li X, He T, Ju F, Qiu Y, Tian Z. Impact of Physical Activity on COVID-19. Int J Environ Res Public Health. 2022;19(21).

  1. Line 386-388: “This observation is likely due to the PA level of this population as CT and PCOV met CDC recommendations for physical activity by the time of the follow-up assessment”. Based on the paper data the Authors could not state this. To demonstrate this causality a proper experiment needs to be run, comparing time without MVPA and time plus MVPA. Please change this sentence, or prove it with consistent literature.

Answer:  Thank you for your comment. We agree with the comment. Please, see highlighted modifications on the manuscript.

Discussion, Paragraph 9

“Although we hypothesize that this observation could be related to the PA levels, our experimental design does not allow us to establish causal effects in this matter. Therefore, we strongly suggest that future studies further investigate this aspect, investigating ANS responses across different ranges of PA, which was not possible in our study due to the small sample size.”

  1. Line 394-397: “Additionally, future studies may also aim to observe how behavioral changes resulting from the COVID-19 395 pandemic have impacted PA behavior and further explore how this may have impact ANS function.” This sentence should be rewritten better.

Answer:  Thank you for your comment. We agree with the comment. We removed this sentence to avoid redundancy with the modifications provided in the previous paragraph (according to comment 20).

  1. Line 406-408: “ANS dysregulation indicates deviation from homeostasis and can effectively predict cardiometabolic illness which is directly correlated with mortality in several diseases, regardless of age (15)”. This sentence is too vague, and the reference discusses mainly cardiac disease. Please be more specific in discussing ANS dysfunction and focus on young adults.

Answer:  Thank you for your comment. We agree with the comment. To provide a better flow in the discussion section we improved the discussion on ANS in young adults and moved this paragraph up in the discussion. Please, see highlighted modifications on the manuscript.

Discussion, paragraph 5

“It is important to highlight that age plays an important role on ANS regulation and our study observed young adults without any known chronic diseases. In young adults, protective factors can be related to better ANS regulation and might help explain the responses of ANS after virus infection observed in our study. Healthy young adults can present lower levels of arterial and ventricular stiffening, better myocardial contractility, and preserved organ innervation when compared to elderly individuals. (46, 47) Additionally, improve sensitivity of vagal reflexes can lead to higher heart rate variability in this population. (48) Therefore, a combination of these aspects could have influenced the improvements observed in PCOV group over time. We believe that our results support the use of HRV monitoring as an important, reliable, and accessible method to observe alterations in autonomic function. Deviations in autonomic regulation are related to distancing from homeostasis and are present in multiple diseases, both those that directly afflict the nervous system as well as those afflicting other organs, where they trigger or enhance pathological symptoms. (15, 49)”

  1. Line 424-426: “The costs associated with equipment and the time for the HRV monitoring in this study are more accessible than other methods, which may work to improve health outcomes after COVID-19” Add references to sustain this statement.

Answer:  Thank you for your comment. We removed this sentence and better clarify the role of HRV in paragraph 5 of the discussion (please refer to comment 22).  

  1. As stated above, the Discussion does not explain the implications of SARS-CoV-2 immunization for ANS balance.

Answer:  Thank you for your comment. We agree with the comment. We included this information as requested. Please, see highlighted modifications on the manuscript.

Discussion, Paragraph 11

“Since we evaluated young adults before and after SARS-CoV-2 immunization, the interpretation of ANS response overtime should also consider the potential effects of the vaccine. Although the time point for testing long-term effect of the vaccine is still relatively short, previous studies comprehensively observed the safety and effectiveness of the COVID‐19 vaccine.(60) Reports on the effects of immunization on ANS regulation are still scarce and isolated. Limited case reports previously presented occurrences of post-COVID-19-vaccine-associated postural orthostatic tachycardia syndrome (POTS), a multifactorial and often debilitating type of autonomic dysfunction.(61-63) These alterations were observed short-term after immunization (4 hours to 7 days) and possible mechanism could be attributed to sinus tachycardia and cerebral hypoperfusion secondary to hypocapnia observed in these cases. This mechanism could result in ischemic hypoxia of the carotid body, leading to chemoreflex activation and sympathetic activation.(61) It is important to highlight that these findings are based on very small samples and the pathophysiological role of the patient’s prior COVID-19 infection still requires further investigation. The reports also state that these observations cannot imply causality of POTS possibly linked to the COVID-19 vaccine. (61)

In our study, the aspect of immunization in our study was used as a controlling factor to avoid bias (all individuals were vaccinated on follow-up). Therefore, the absence of a non-vaccinated group in our design limits our ability to differentiate and account for the specific effects of immunization on ANS. Due to ethical aspects and the high rates of immunization after the availability of immunization in Brazil (over 88% of the population received at least one shot of immunization), it was difficult to incorporate a non-vaccinated group. Future investigations should consider the investigation of ANS function in non-vaccinated individuals to further advance knowledge in this aspect.”

  1. In addition, I report below some grammatical errors in the Discussion and Conclusion sections that should be reviewed.

-       Line 298: change “6-weeks” in “6 weeks”

-       Lines 301,304, 430: change “follow up” to “follow-up”

-       Line 317: change “cross sectional” to “cross-sectional”

-       Line 343: change “increase” to “increases”

-       Line 405: “as evidence as to the use”. Change prepositions to make clearer the whole sentence

-       Line 412: change “cumbersome on both” to “cumbersome for both”

-       Line 422: change “of monitoring of ANS” to “of monitoring ANS”

-       Line 429: change “5 month” to 5-month”

Answer:  Thank you for your comment. Please, see highlighted modifications on the manuscript.

*Please see full manuscript attached.

Reviewer 2 Report

In this article the Authors focus on long-term recovery of young adults after mild-to-moderate COVID-19. They study the effects of COVID-19 on autonomic nervous system (ANS) measuring heart rate variability (HRV) as a surrogate of ANS activity. They also focus on physical activity (PA) in its role to improve ANS modulation.

I think that this interesting paper tries to fill the gap between scientific evidence of long COVID syndrome and objective non-invasive measurement of ANS modulation.

However, I think that there are some areas of weakness that the Authors should consider.

First, the subjects (PCOV) recruited for the study at baseline were tested negative for SarS-CoV2 after a wide range of days - 15 up to 120 days. These subjects have had mild or moderate COVID-19. This may be a limitation because some PCOV subjects may have experienced symptoms such as chest pain or severe myalgias indicative of myocarditis or pericarditis, while still not requiring intensive care. For evaluating HRV, it is important to consider the possible burden of premature ventricular contractions (PVC), which may be triggered by myo-pericarditis. I think that a more specific definition of mild and moderate COVID-19 should be given.

Second, I would ask why the Authors chose to measure HRV in a sitting position for 25 minutes, and to cite a reference for this method; it seems more usual to measure HRV on supine position from Literature. Regarding the measures of HRV, I would also ask if the Authors considered very low frequency (VLF) power, since it appears to be influenced by renin-angiotensin system, which is known to be the target of SarS-CoV2.

Third, I really appreciated that physical activity (PA) was objectively measured the week before each evaluation via accelerometers, but it should be specified whether the subjects recruited followed or not a rehabilitation program between the evaluations or were engaged in regular sports activity (and maybe overtraining?).

Fourth, HRV could be affected by mental health (i.e., depression); have the Authors studied this aspect in their subjects, as it may alter HRV measures?

Finally, the Authors should consider rewording of the conclusion (line 430), because it seems to conflict with lines 334-338.

Please also check:

-       Lines 130,131: the references do not match with physical activity level and BMI

-       Line 137: number of days

-       Line 140: definition of BMI

-       Table 2: the line of weight and height are mismatched

Author Response

REVIEWER 2

In this article the Authors focus on long-term recovery of young adults after mild-to-moderate COVID-19. They study the effects of COVID-19 on autonomic nervous system (ANS) measuring heart rate variability (HRV) as a surrogate of ANS activity. They also focus on physical activity (PA) in its role to improve ANS modulation. I think that this interesting paper tries to fill the gap between scientific evidence of long COVID syndrome and objective non-invasive measurement of ANS modulation. However, I think that there are some areas of weakness that the Authors should consider.

  1. First, the subjects (PCOV) recruited for the study at baseline were tested negative for SarS-CoV2 after a wide range of days - 15 up to 120 days. These subjects have had mild or moderate COVID-19. This may be a limitation because some PCOV subjects may have experienced symptoms such as chest pain or severe myalgias indicative of myocarditis or pericarditis, while still not requiring intensive care.

Answer:  Thank you for your comment. We agree with the comment. We added this aspect as a limitation of the study. Please, see highlighted modifications on the manuscript.

Discussion, Paragraph 14

“Last, although we excluded individuals that required intensive care from our sample, subjects may have experienced symptoms such as chest pain or severe myalgias indicative of myocarditis or pericarditis, while still not requiring intensive care. This aspect was not controlled in our study and needs to be considered for the interpretation of our results.”

  1. For evaluating HRV, it is important to consider the possible burden of premature ventricular contractions (PVC), which may be triggered by myo-pericarditis. I think that a more specific definition of mild and moderate COVID-19 should be given.

Answer:  Thank you for your comment. We agree with the comment. We included this definition in the inclusion and exclusion criteria. Please, see highlighted modifications on the manuscript.

Methods, Paragraph 4

“Inclusion criteria were male and female subjects aged 20-40 years with, a diagnosis of mild or moderate clinical COVID-19 with a previous positive polymerase chain reaction (PCR) test. Mild cases were considered when patients presented slight clinical symptoms without imaging findings of pneumonia. Moderate cases were considered when patients reported fever or respiratory symptoms.(29) Participants were recruited after a minimum of 15 and a maximum of 120 days of diagnosis by positive PCR test (30). An age-matched healthy control group (CT) that was negative for COVID-19 was also recruited. To screen for confirmed or probable previous SARS-CoV-2 infection for the control group, a lateral flow test for SARS-CoV-2 Immunoglobin G (IgG) and Immunoglobin M (IgM) antibodies was conducted using amplified chemiluminescence and chemiluminescence serological methods, respectively. Subjects in both groups were eligible for a follow-up evaluation if they received any type SARS-CoV-2 immunization after initial evaluations.

            We excluded subjects that presented severe clinical manifestations of COVID-19, including respiratory distress and a respiratory rate >30 times per minute, fingertip blood oxygen saturation <93% at rest, and partial arterial oxygen pressure (PaO2)/fraction of inspiration oxygen (FiO2) <300 mmHg. Subjects that reported critical conditions including respiratory failure requiring mechanical ventilation, shock, and other organ failure or requiring ICU treatment were also excluded.(29)”

  1. Second, I would ask why the Authors chose to measure HRV in a sitting position for 25 minutes and to cite a reference for this method; it seems more usual to measure HRV on supine position from Literature.

Answer:  Thank you for your comment. Previous literature supports HRV data collection while sitting, especially with studies focusing on cardiac vagal tone. Please see the study of Laborde et al. (Heart Rate Variability and Cardiac Vagal Tone in Psychophysiological Research – Recommendations for Experiment Planning, Data Analysis, and Data Reporting). We added this reference to the manuscript. Additionally, the Task Force of the European Society of Cardiology recommendation is to standardize measurements, making assessments as consistent as possible. Following these recommendations, we were able to keep subjects in the same position for all evaluations and reduce the probability of patients following asleep during the assessment, which could also alter HRV results.

Regarding the duration of monitoring, the Task Force of the European Society of Cardiology recommendation is to perform recordings of at least 5 minutes under physiologically stable conditions, especially for processing frequency and time domain methods. We excluded the first and last 5 to 8 minutes of monitoring to select the most stable section of the analysis.

Reference: Heart rate variability: standards of measurement, physiological interpretation and clinical use. Task Force of the European Society of Cardiology and the North American Society of Pacing and Electrophysiology. Circulation. 1996;93(5):1043-65.

  1. Regarding the measures of HRV, I would also ask if the Authors considered very low frequency (VLF) power, since it appears to be influenced by renin-angiotensin system, which is known to be the target of SarS-CoV2.

Answer:  Thank you for your comment. Although we did consider VLF as an important measurement, we chose to reduce the number of indexes included for HRV analysis and avoid multiple comparisons that could lead to Type I error.

  1. Third, I really appreciated that physical activity (PA) was objectively measured the week before each evaluation via accelerometers, but it should be specified whether the subjects recruited followed or not a rehabilitation program between the evaluations or were engaged in regular sports activity (and maybe overtraining?).

Answer:  Thank you for your comment. None of the participants participated in rehabilitation programs. This information was included in the exclusion criteria. Regarding sports activity, we included individuals with all levels of physical activity and performed the analysis of covariance (ANCOVA) to avoid confusing influences of PA on HRV indexes (please see Table 3)

  1. Fourth, HRV could be affected by mental health (i.e., depression); have the Authors studied this aspect in their subjects, as it may alter HRV measures?

Answer:  Thank you for your comment. Yes, anxiety, depression and sleep were measured in these subjects but the reporting on these outcomes will be included in a different separate study.

  1. Finally, the Authors should consider rewording the conclusion (line 430), because it seems to conflict with lines 334-338.

Answer:  Thank you for your comment. Please, see highlighted modifications on the manuscript.

Conclusions

We observed improvements in ANS function after mild-to-moderate SARS-CoV-2 infection in young adults vaccinated over 5 months. Additionally, no changes in physical activity levels were observed over the follow-up period.”

  1. Please also check:

-       Lines 130,131: the references do not match with physical activity level and BMI

-       Line 137: number of days

-       Line 140: definition of BMI

-       Table 2: the line of weight and height are mismatched

Answer:  Thank you for your comment. Please, see highlighted modifications on the manuscript.